# The prevalence and correlates of depression among patients with chronic diseases in the United Arab Emirates

**A. J. Alkaabi**[1], **A. Alkous**[1], **K. Mahmoud**[1], **A. AlMansoori**[1], **Iffat Elbarazi**[1], **Abubaker Suliman**[1], **Zufishan Alam**[1], **Fatheya AlAwadi**[2], **Fatima Al-Maskari**[1,3] *

**1** Institute of Public Health, College of Medicine and Health Sciences, United Arab Emirates University, AlAin, Abu Dhabi, UAE, **2** Ambulatory Health Services, SEHA, Abu Dhabi, UAE, **3** Zayed Centre for Health Sciences, United Arab Emirates University, AlAin, Abu Dhabi, UAE

* fatma.am@uaeu.ac.ae

## Abstract

### Background

Chronic diseases constitute a major public health problem in the United Arab Emirates (UAE) and are the leading cause of mortality and morbidity. Chronic diseases have been found to be associated with an increased prevalence of depression and depressive symptoms. Depression can have detrimental effect on the prognosis of the disease and quality of life in patients.

### Aims and objectives

This study aimed to estimate the prevalence and correlates of depression in a sample of patients suffering from chronic disease in Al-Ain city, UAE.

### Materials and methods

A cross-sectional survey based study was conducted with 417 participants recruited from seven primary health care centers of Al-Ain city. Men and women aged 18 years and above suffering from chronic disease filled the Patient Health Questionnaire (PHQ-9). Univariate and multivariable logistic regressions were performed on the collected data to investigate correlates of different factors with depression. Data was analyzed using SPSS (version 26). The study was approved by Ambulatory Healthcare Services (AHS) Human Ethics Research Committee.

### Results

The majority 62.41% (n = 254) of the sample were females, 57.97% (n = 240) aged above 55 years and with a median (Q25, Q75) duration of chronic disease of 8 (4, 15) years. The prevalence of depression was 21.1% (95% CI: 17.5%–25.3%). With severe depression was in 1.7% and mild-moderate in 34.7% of the participants. Depression severity was statistically significantly associated with increasing age (p = 0.006), low level of education (p<0.001), presence of asthma (p = 0.007) and heart disease (p = 0.013). Unadjusted

**Data Availability Statement:** All data files are available from Harvard Dataverse public repository (https://doi.org/10.7910/DVN/NRJNEN).

**Funding:** This study was funded by SUREPLUS Grant, G00003183 from United Arab Emirates University. The funders had no role in study design, data collection and analysis, decision to publish, or preparation of the manuscript.

**Competing interests:** The authors have declared that no competing interests exist.

logistic regression reported that presence of depression was significantly associated with female gender (cOR = 1.8, [95% CI; 1.1–3.1], $p$ = 0.025), and presence of chronic kidney disease (cOR = 4.9, [95% CI; 1.3–20.2], $p$ = 0.020) and heart disease (cOR = 2.9, [95% CI; 1.6–5.4], $p$ = 0.001) longer duration of disease in years (cOR = 1.04, [95% CI; 1.01–1.07], $p$ = 0.003). However, in the adjusted logistic regression analysis, participants with heart disease (aOR = 2.8, [95% CI; 1.4–5.5], $p$ = 0.004), and with longer duration of disease (aOR = 1.04, [1.01–1.07], $p$ = 0.014) remained significantly associated statistically with higher chance of having depression.

## Conclusion

The prevalence of depression was quite high and the study highlights for health care professionals and policy makers, the importance of mental health support as part of a comprehensive management plan for patients with chronic diseases. A multidisciplinary comprehensive program will improve the long-term outcomes of these patients. Patients with chronic diseases may need more support and counseling at primary health care levels.

## Introduction

Depression also known as "major depressive disorder" is an ailment that affects an individuals' perception of how they feel, think or act. It can be the reason for sadness or lack of interest in activities among these individuals. Depression may present with physical symptoms, fatigue, pain, or sleep disturbances [1]. It is estimated to affect 3.8% of the total world population, of which 5.0% are adults and 5.7% are adults aged 60 and above [2]. Depression can be a leading cause of suicide and 700,000 people die annually due to suicide. Even though there are effective treatments to cure depression, more than 75% of people from low- and -middle-income countries have no access to such treatments [3].

Enduring a physical illness is one of the strongest risk factors for depression. In a study by Clarke & Currie, (2009), it was reported that prevalence of depression was significantly and constantly greater in patients suffering from chronic diseases such as heart diseases, diabetes mellitus, cancer, rheumatoid arthritis as compared to that in overall population [3]. The CDC defines a chronic disease as "conditions that last 1 year or more and require ongoing medical attention or limit activities of daily living or both". Chronic diseases such as heart disease, cancer, and diabetes [4]. Patients with chronic diseases are more prone to have depression but its overlapping clinical symptoms makes it harder to be diagnosed. In fact, undiagnosed depression is becoming a major concern in primary care [5]. Mutual risk factors and pathophysiological routes strongly associate depression with chronic diseases. Lotfaliany et al. (2018) found that most of the depression cases were not diagnosed properly in low- and middle-income countries where a positive relation exists between undiagnosed depression and higher risk of suffering from diabetes, arthritis, asthma, and stroke [6]. Furthermore, one in four diabetic patients presents with depression during a time period of 2.5 years and it, thus commonly becomes a chronic condition among such patients. Depression is further associated with a 60% higher risk of developing type 2 diabetes [7].

Unfortunately, more than 51 million people die annually from a non-communicable disease (NCD) and 77 percent of NCDs occur in low and low-middle income countries [8]. In the United Arab Emirates (UAE), depression is among the top three causes of disability-adjusted

life years [9]. The most prominent risk factors for depression in the UAE are family history of chronic diseases, female gender, lower socio-economic status and absence of social support. Moreover, the burden of chronic diseases in this region is considerably high. In UAE, in 2010, 27% of deaths were attributed to diseases of the cardiovascular system even within its younger population and prevalence rate for diabetes was second highest around the globe [10]. However, there is still a gap in the literature on the correlation between depression and suffering from a chronic disease in the UAE. New research is needed in this area to cover up existing gaps in the literature. This study aimed to identify the prevalence and risk factors for depression among patients with chronic illnesses in the UAE in order to improve the awareness and management of depression for better quality of life.

## Materials and methods

This cross-sectional study was conducted in seven Ambulatory Health Services (AHS) centers in Al Ain city, the largest city in Abu Dhabi Emirate, between the periods of June 2020 and December 2020. Prior to starting the study, a co-investigator from AHS was contacted to assist in facilitation of the study process and ethics approval from AHS Ethics Committee was obtained. We have chosen the AHS clinics as being the best setting for the study for two main reasons. The first accessibility to the target group as some of the AHS clinics in Al Ain city have special clinics for patients with chronic diseases called (Chronic Disease clinic–(CDC)). The second reason these clinics provide the service of diagnosis, treatment, and follow-up with patients with chronic diseases. Most of the patients who are seen in the CDC clinic come frequently for medication refill, for regular checkup without specific acute complains and for prevention of complication. Moreover, it is a common practice by health care team in these clinics to ask patients to fill in the PHQ9 however, the prevalence and the mental health needs of these patients never been studied based on the results of the PHQ9 in these clinics. The study was conducted in accordance with the Declaration of Helsinki and approved by the Ambulatory Healthcare Services (AHS) Human Ethics Research Committee. Written Informed consent was obtained from all subjects involved in the study.

### Participants and sample size

Patients above 18 years of age, of any gender and nationality and those suffering from one chronic disease or more such as diabetes, cancer, cardiovascular diseases, kidney diseases, auto immune disease and other chronic diseases as per CDC definition, were invited to participate. Patients with medical psychiatric comorbidities or previously diagnosed depression, and those unable to consent were excluded.

We assumed that the expected prevalence of depression among chronically diseased population in the UAE would be 12.5% based on a previous study that estimated prevalence of depression among patients with Type 2 Diabetes in UAE [11]. Using epi- tools we calculated our sample based on the prevalence of 12.5%, confidence interval of 95% and a confidence level of 5%. A sample of 196 was needed. Expecting that 50% of our sample will be non-responders a sample of 392 participants were required to estimate the prevalence of depression among UAE chronic disease patients.

Participants were approached through reception and waiting areas of health care centers and presented with the invitation letter, consent form and questionnaire. A box was provided on site to return the filled in questionnaire for those who approved and filled in. Whereas the participants not able to fill the questionnaire were assisted by the Research Assistant and medical students. All participants were prompted to consent in writing when approached by nurses

and data collectors to fill in the questionnaire. Only those who consented were given the questionnaire to fill it and were advised to return it to the box.

### Data collection and survey tool

The survey tool consisted of a self-administered questionnaire consisting of a demographic section and a Patient Health Questionnaire (PHQ9). The tool was developed by medical students and a Research Assistant and piloted among 25 patients to identify any gaps and for scope and clarity. It was then distributed in randomly chosen seven AHS centers in Al Ain city.

PHQ-9, the instrument used in the survey, is validated and has been widely used in literature. It was developed and validated in 2001 [12] and consists of nine items to assess prevalence of depression in an individual. The score for each question varies from 0 to 3 (0 = not at all, 1 = several days, 2 = more than half of the days, 3 = nearly every day), with a result range of 0–27. score of 0–4 indicates 'minimal depression', 5–9 indicates 'mild depression', 10–14 indicates 'moderate depression', 15–19 indicates 'moderately severe depression' and 20–27 indicates 'severe depression'. Its score of $\geq$10 was shown to have a sensitivity of 88% and a specificity of 88% for major depression. PHQ-9 scores of 5, 10, 15, and 20 represent mild, moderate, moderately severe, and severe depression, respectively for the current study, the scores were re-categorized as follows: minimal depression (score 0–4), mild depression (score 5–14) and moderate depression (score 15–27). The recategorization was adapted from other studies [13–15]. A cut off score of 8 was chosen for the diagnosis of depression based on earlier meta-analysis [16].

### Statistical analysis

Descriptive analysis was performed, with categorical variables presented using frequencies and percentages, while continuous variable summarized using median (Q25, Q75). The Chi-square or Fisher's exact test (categorical variables), and the Kruskal-Wallis test (continuous variable) were used to compare demographic and comorbid characteristics between depression categories. To assess correlates of depression with different factors, we first fitted simple logistic regressions with depression (PHQ-9 score $\geq$ 8) as the dependent variable and gender, age, work status, education, type I diabetes, type II diabetes, hypertension, heart disease, chronic kidney disease, asthma, duration of disease in years, as independent variables. Second, a multiple logistic regression was used to examine the variables independently associated with depression. The multiple logistic regression included all independent variables with a P value < 0.10 in the unadjusted analysis. All P values were 2-sided and P < 0.05 was considered a statistically significant. All analyses were conducted using SPSS (version 26) [17].

### Results

Of the 417 participants, majority 62.41% (n = 254) were females, 21.52% (n = 88) with university level education, 57.97% (n = 240) aged above 55 years and with a median (Q25, Q75) duration of chronic disease of 8 (4, 15) years (Table 1). The prevalence of depression was 21.1% (95% CI: 17.5%–25.3%) based on a cut-off score of 8. Severe depression was present in 1.7% and mild-moderate in 34.7% of the participants (Fig 1). Depression prevalence was higher in patients with heart disease (40.0%) compared to those with hypertension (21.5%) (Fig 2).

Association between depression severity and different demographic and comorbidities characteristics was presented in Table 2. Depression severity was significantly associated with increasing age (p = 0.006), low level of education (p<0.001), presence of asthma (p = 0.007)

**Table 1. Demographic and comorbidities characteristics of patients with chronic disease in the UAE (N = 417).**

| Variable | Statistic |
|---|---|
| **Gender, n (%)** | |
| Male | 153 (37.59) |
| Female | 254 (62.41) |
| **Age, n (%)** | |
| 20–34 years | 36 (8.70) |
| 35–54 years | 138 (33.33) |
| ≥55 years | 240 (57.97) |
| **Employment status, n (%)** | |
| Unemployed | 296 (70.98) |
| Employed full or part time | 113 (27.10) |
| **Education, n (%)** | |
| Illiterate | 115 (28.12) |
| School degree (primary, preparatory, secondary) | 206 (50.37) |
| University degree | 88 (21.10) |
| **Type I diabetes, n (%)** | |
| No | 394 (94.48) |
| Yes | 23 (5.52) |
| **Type II diabetes, n (%)** | |
| No | 197 (47.24) |
| Yes | 220 (52.76) |
| **Hypertension, n (%)** | |
| No | 175 (41.97) |
| Yes | 242 (58.03) |
| **Heart disease, n (%)** | |
| No | 367 (88.01) |
| Yes | 50 (11.99) |
| **Chronic kidney disease, n (%)** | |
| No | 408 (97.84) |
| Yes | 9 (2.16) |
| **Cancer, n (%)** | |
| No | 413 (99.04) |
| Yes | 4 (0.96) |
| **Asthma, n (%)** | |
| No | 381 (91.37) |
| Yes | 36 (8.63) |
| **Autoimmune, n (%)** | |
| No | 414 (99.28) |
| Yes | 3 (0.72) |
| **Years with condition, Median (Q25, Q75)** | 8 (4, 15) |

and heart disease (p = 0.013). We found no association between gender, employment status and the duration of chronic disease with depression severity.

The unadjusted logistic regression analysis indicated that being a female (crude odds ratio; cOR = 1.8, [95% CI; 1.1–3.1], p = 0.025), having heart disease (cOR = 2.9, [1.6–5.4], p = 0.001), having chronic kidney disease (cOR = 4.9, [1.3–20.2], p = 0.020), having the chronic disease for longer duration in years (cOR = 1.04, [1.01–1.07], p = 0.003) were statistically significant associated with higher odds of having depression. However, as shown form the adjusted

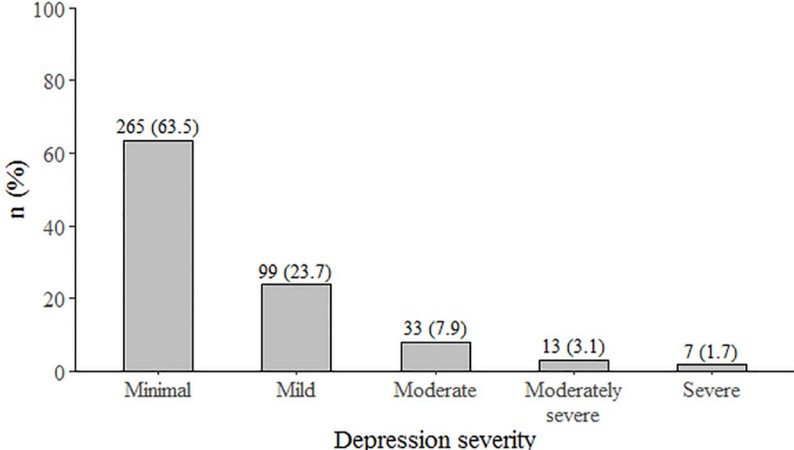

**Fig 1. Distribution of depression severity in patients with chronic disease in the UAE (N = 417).**

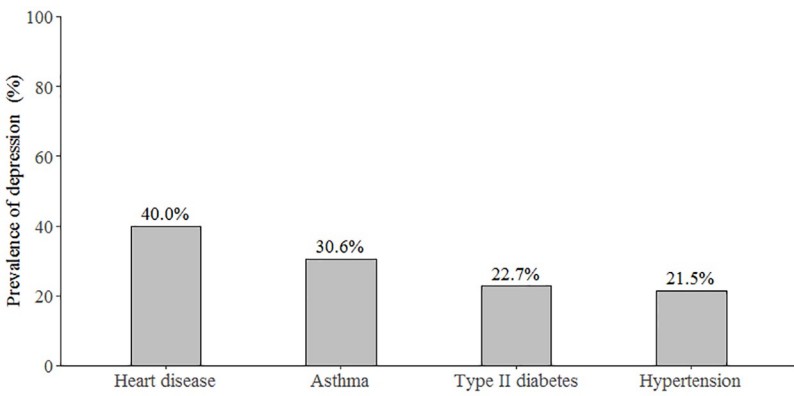

**Fig 2. Depression prevalence in selected chronic diseases.**

logistic model, only heart disease presence (adjusted odds ratio; aOR = 2.8, [95% CI; 1.4–5.5], p = 0.004), and longer duration of chronic disease in years (aOR = 1.04, [1.01–1.07], p = 0.014) were independently associated with depression (Table 3).

## Discussion

This study assessed depression prevalence and associated factors among patients suffering from chronic disease in the UAE. Main findings indicate that while 34.7% of the sample had mild to moderate depression, only 1.7% reported having severe depression in the survey. Being a female, having a heart disease, having a chronic kidney disease, and longer duration of having the chronic disease were significantly associated statistically with higher odds of depression in the univariate regression, and heart disease and chronic disease duration in multivariate regression.

The results from this study confirm to the findings from another multistage, stratified, cross-sectional population survey-based study, carried out in Dubai, where prevalence of depressive disorders among adult population was found to be significantly associated with pre-existing health conditions such as stroke, chest pain and tuberculosis [11]. Similarly, few

**Table 2. Association of depression severity with demographic and comorbidities characteristics of patients with chronic disease in the UAE (N = 417).**

| Variable | Severity of depression, N (%) | | | p-value [a] |
|---|---|---|---|---|
| | Minimal 364 (87) | Moderate 33 (8) | Severe 20 (5) | |
| **Gender, n (%)** | | | | |
| Male | 138 (38.98) | 8 (24.24) | 7 (35.00) | 0.240 |
| Female | 216 (61.02) | 25 (75.76) | 13 (65.00) | |
| **Age, n (%)** | | | | |
| 20–34 years | 28 (7.76) | 3 (9.09) | 5 (25.00) | 0.006 |
| 35–54 years | 127 (35.18) | 11 (33.33) | 0 (0.00) | |
| ≥55 years | 206 (57.06) | 19 (57.58) | 15 (75.00) | |
| **Employment status, n (%)** | | | | |
| Unemployed | 252 (69.23) | 27 (81.82) | 17 (85.00) | 0.172 |
| Employed full or part time | 104 (28.57) | 6 (18.18) | 3 (15.00) | |
| **Education, n (%)** | | | | |
| Illiterate | 91 (25.56) | 10 (30.30) | 14 (70.00) | < 0.001 |
| School degree | 190 (53.37) | 14 (42.42) | 2 (10.00) | |
| University degree | 75 (20.60) | 9 (27.27) | 4 (20.00) | |
| **Type I diabetes, n (%)** | | | | |
| No | 342 (93.96) | 32 (96.97) | 20 (100.00) | 0.684 |
| Yes | 22 (6.04) | 1 (3.03) | 0 (0.00) | |
| **Type II diabetes, n (%)** | | | | |
| No | 170 (46.70) | 18 (54.55) | 9 (45.00) | 0.674 |
| Yes | 194 (53.30) | 15 (45.45) | 11 (55.00) | |
| **Hypertension, n (%)** | | | | |
| No | 151 (41.48) | 16 (48.48) | 8 (40.00) | 0.725 |
| Yes | 213 (58.52) | 17 (51.52) | 12 (60.00) | |
| **Heart disease, n (%)** | | | | |
| No | 325 (89.29) | 29 (87.88) | 13 (65.00) | 0.013 |
| Yes | 39 (10.71) | 4 (12.12) | 7 (35.00) | |
| **Chronic kidney disease, n (%)** | | | | |
| No | 358 (98.35) | 32 (96.97) | 18 (90.00) | 0.059 |
| Yes | 6 (1.65) | 1 (3.03) | 2 (10.00) | |
| **Cancer, n (%)** | | | | |
| No | 361 (99.18) | 32 (96.97) | 20 (100.00) | 0.421 |
| Yes | 3 (0.82) | 1 (3.03) | 0 (0.00) | |
| **Asthma, n (%)** | | | | |
| No | 337 (92.58) | 30 (90.91) | 14 (70.00) | 0.007 |
| Yes | 27 (7.42) | 3 (9.09) | 6 (30.00) | |
| **Autoimmune, n (%)** | | | | |
| No | 361 (99.18) | 33 (100.00) | 20 (100.00) | 1.000 |
| Yes | 3 (0.82) | 0 (0.00) | 0 (0.00) | |
| **Years with condition, Median (Q25, Q75)** | 8 (4, 15) | 12 (5, 20) | 8 (5, 14) | 0.249 |

[a] Categorical variables analyzed using Chi-square or Fisher's exact test and continuous variable by Kruskal-Wallis test.

previous studies that used PHQ-9 to assess depression prevalence among patients suffering from chronic diseases such as multiple sclerosis and epilepsy in the UAE, have reported it to be more prevalent (17% and 27% respectively) than in healthy population [18, 19]. Another study carried out in 2014 to assess depression in patients suffering from NCDs and attending

**Table 3. Simple and multiple logistic regression analysis of factors affecting depression in chronic disease patients in the UAE (N = 417).**

| Variable | cOR (95% CI) | p-value | aOR (95% CI) | p-value |
|---|---|---|---|---|
| **Gender** | | | | |
| Male | Ref | - | Ref | - |
| Female | 1.8 (1.1–3.1) | 0.025 | 1.9 (1.0–3.6) | 0.054 |
| **Age** | | | | |
| 20–34 years | Ref | - | Ref | - |
| 35–54 years | 0.4 (0.2–1.1) | 0.065 | 0.3 (0.1–0.9) | 0.024 |
| > = 55 years | 0.8 (0.4–1.9) | 0.640 | 0.5 (0.2–1.4) | 0.165 |
| **Employment status** | | | | |
| Unemployed | Ref | - | Ref | - |
| Employed full or part time | 0.6 (0.3–1.0) | 0.051 | 0.8 (0.4–1.7) | 0.536 |
| **Education** | | | | |
| Illiterate | Ref | - | Ref | - |
| School degree | 0.5 (0.3–0.8) | 0.009 | 0.6 (0.3–1.1) | 0.118 |
| University degree | 0.6 (0.3–1.2) | 0.142 | 0.9 (0.4–2.3) | 0.853 |
| **Type I diabetes** | | | | |
| No | Ref | - | - | - |
| Yes | 0.8 (0.2–2.1) | 0.654 | - | - |
| **Type II diabetes** | | | | |
| No | Ref | - | - | - |
| Yes | 1.2 (0.8–2.0) | 0.391 | - | - |
| **Hypertension** | | | | |
| No | Ref | - | - | - |
| Yes | 1.1 (0.7–1.7) | 0.821 | - | - |
| **Heart disease** | | | | |
| No | Ref | - | Ref | - |
| Yes | 2.9 (1.6–5.4) | 0.001 | 2.8 (1.4–5.5) | 0.004 |
| **Chronic kidney disease** | | | | |
| No | Ref | - | Ref | - |
| Yes | 4.9 (1.3–20.2) | 0.020 | 2.3 (0.5–10.4) | 0.271 |
| **Asthma** | | | | |
| No | Ref | - | - | - |
| Yes | 1.7 (0.8–3.6) | 0.150 | - | - |
| **Years with condition** | 1.04 (1.01–1.07) | 0.003 | 1.04 (1.01–1.07) | 0.014 |

cOR: Crude Odds ratio, aOR: Adjusted Odds ratio, CI: Confidence Interval

primary health care centres in Dubai UAE reported that nearly 33% had mild-moderate depression similar to results from this study [20].

Various studies have been carried out around the world to estimate depression among patients with chronic diseases. A study using data from WHO surveys for assessment of depression in chronically diseased patients in 64 countries, reported it to be between 9.3% -23.0%. It also reported that depression in combination with chronic disease had significant effect on worsening health scores [21]. Similarly another review has established the link between depression and chronic disease, emphasizing the role of proper care management [22]. Evaluation of literature on multimorbidity caused by NCDs in WHO Eastern Mediterranean region

including Syria, Jordan, Lebanon, and Qatar has indicated that depression is consistently prevalent among patients with chronic disease [23].

However, an important point to take under consideration is that studies have also indicated the possibility of depression, leading to chronic disease development. A Canadian population-based cohort study concluded that major depression is significantly associated with long term disease and therefore should be considered as a risk factor for chronic disease development [24]. Likewise, a central theme outlined in qualitative meta synthesis of studies, carried out on patients suffering from chronic disease and depression, on relationship of both conditions, was emergence of chronic disease following depression [25]. Thus the importance of timely diagnosis of depression cannot be underestimated.

The result from this study, indicating that longer duration of disease is associated with depression, has been reported previously as well. A recent large Chinese cohort study indicated increased chance for having depression with course of disease being more than 5 years [26]. Similarly, association of female gender with depression in chronic patients has also been reported [27]. Various studies support our finding that patients suffering from heart disease and chronic kidney disease area at more risk of having depression, thus adversely affecting quality of life [28].

Health care workers are faced with a significant challenge when patients have a chronic medical disease and a mental disorder, presenting with psychological symptoms and a need for demanding treatment regimen. Role of tailored, multidisciplinary management in treatment of depression among chronic disease patients, has been investigated previously, indicating its significance. Literature evaluation of various randomized controlled trials for chronic disease management with depression, provides evidence on the role of incorporation of multiple components in primary care practices to ensure proper organization and delivery of care services. Various components outlined include incorporation of help from mental health professionals, regular follow up, and self-care management support [29]. Stress has been placed to include both biomedical and psychological aspects of care delivery. An potential example of intervention developed based on the principle could be enabling and training health professionals such as nurses in behavioral management [30].

This study was the first to evaluate depression prevalence and associated factors within a large study sample of patients suffering from chronic diseases in Al Ain. Since the response to the survey was self-reported, a limitation of the study could be introduction of social desirability bias. However other studies carried out in Arab countries have reported similar prevalence level, thus rendering the results to be true. Nevertheless, this study affirms to findings from wider literature that depression is more common among people with chronic diseases. The significant association of heart disease and longer time of suffering with the disease emphasizes that care for such patients need to include relevant and tailored intervention programs to support mental health, as much as physical health.

## Acknowledgments

We would like to acknowledge SEHA-AHS centers for providing access for data collection.

## Author Contributions

**Conceptualization:** A. J. Alkaabi, A. Alkous, K. Mahmoud, A. AlMansoori, Iffat Elbarazi, Abubaker Suliman, Fatima Al-Maskari.

**Data curation:** A. J. Alkaabi, A. Alkous, K. Mahmoud, A. AlMansoori, Iffat Elbarazi, Fatheya AlAwadi.

**Formal analysis:** Abubaker Suliman.

**Funding acquisition:** Iffat Elbarazi, Fatima Al-Maskari.

**Investigation:** Iffat Elbarazi.

**Methodology:** Iffat Elbarazi, Fatima Al-Maskari.

**Project administration:** Fatheya AlAwadi.

**Resources:** Fatima Al-Maskari.

**Supervision:** Iffat Elbarazi, Fatheya AlAwadi, Fatima Al-Maskari.

**Validation:** Iffat Elbarazi.

**Visualization:** Abubaker Suliman.

**Writing – original draft:** A. J. Alkaabi, A. Alkous, K. Mahmoud, A. AlMansoori, Iffat Elbarazi, Abubaker Suliman, Zufishan Alam, Fatheya AlAwadi, Fatima Al-Maskari.

**Writing – review & editing:** Iffat Elbarazi, Abubaker Suliman, Zufishan Alam, Fatima Al-Maskari.

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
