## [Decision Letter · Decision Letter 0]

28 Oct 2022

PONE-D-22-26056The Prevalence and Correlates of Depression among Patients with Chronic Diseases in the United Arab EmiratesPLOS ONE

Dear Dr. Suliman,

Thank you for submitting your manuscript to PLOS ONE. After careful consideration, we feel that it has merit but does not fully meet PLOS ONE’s publication criteria as it currently stands. Therefore, we invite you to submit a revised version of the manuscript that addresses the points raised during the review process. Please clarify the definitions of each chronic disease investigated in the manuscript. If needed, please also include a section for inclusion and exclusion criteria. A reviewer found the analysis presented in table 2 a bit confusing. Please clarify. The authors have suggested in the paragraph following table 2 that the authors are describing a univariate result analysis. This seems to be the results of a bivariate analysis with Crude Odds Ratios presented. Please clarify. A reviewer also wondered if there was any rational for choosing the Ambulatory Health Services for the recruitment of patients, knowing that patients who present to such services usually come in an acute state. This might influence the way participants respond to the questions on the PHQ questionnaire. Recruiting patients in a more stable environment might give a totally different results.

Please submit your revised manuscript by Dec 12 2022 11:59PM. If you will need more time than this to complete your revisions, please reply to this message or contact the journal office at plosone@plos.org. Please include the following items when submitting your revised manuscript:A rebuttal letter that responds to each point raised by the academic editor and reviewer(s). You should upload this letter as a separate file labeled 'Response to Reviewers'.A marked-up copy of your manuscript that highlights changes made to the original version. You should upload this as a separate file labeled 'Revised Manuscript with Track Changes'.An unmarked version of your revised paper without tracked changes. You should upload this as a separate file labeled 'Manuscript'.

We look forward to receiving your revised manuscript.

Kind regards,

Mohsen Abbasi-Kangevari

Academic Editor

PLOS ONE

**Journal Requirements:**

"This study was funded by SUREPLUS Grant, G00003183 from United Arab Emirates University."

"This study was funded by SUREPLUS Grant, G00003183 from United Arab Emirates University. The funders had no role in study design, data collection and analysis, decision to publish, or preparation of the manuscript."

Reviewers' comments:

Reviewer's Responses to Questions

**Comments to the Author**

1. Is the manuscript technically sound, and do the data support the conclusions?

Reviewer #1: Yes

Reviewer #2: No

2. Has the statistical analysis been performed appropriately and rigorously? 

Reviewer #1: Yes

Reviewer #2: Yes

3. Have the authors made all data underlying the findings in their manuscript fully available?

Reviewer #1: Yes

Reviewer #2: No

4. Is the manuscript presented in an intelligible fashion and written in standard English?

Reviewer #1: Yes

Reviewer #2: Yes

5. Review Comments to the Author

Reviewer #1: This is a cross-sectional study to examine the prevalence and corelates of depression among a sample of patients with chronic diseases in the UAE. The study is well conducted and presented. I would suggest to scale the y-axis of figure 1 from 0-100.

Reviewer #2: It was a pleasure to review this piece of work. I must say this is an interesting piece of work.

I just have a few comments to make.

1. The article is entitled 'The Prevalence and Correlates of Depression among Patients with Chronic Diseases in the United Arab Emirates'. This suggests that all participants recruited have a chronic medical condition. In the manuscript, I couldn't find anywhere the definition for what was considered a chronic medical condition and how this guided the authors in choosing their participants. I would suggest including a section for inclusion and exclusion criteria. This will make it clearer for the readers.

2. I also found the analysis presented in table 2 a bit confusing, can the authors confirm that everyone who was included in the study had a chronic medical condition? If that was the case then our input variable can be broadly classified into 'present or absent' of a chronic medical condition and the outcome variables will be the correlates or associated factors. A separate table may be used to show the prevalence of depression in the different categories of chronic medical condition present in the participants

3. The authors have suggested in the paragraph following table 2 that the authors are describing a univariate result analysis. This seems to be the results of a bivariate analysis with Crude Odds Ratios presented. Can the authors verify this.

4. I was wondering if there was any rational for choosing the Ambulatory Health Services for the recruitment of patients, knowing that patients who present to such services usually come in an acute state. This might influence the way participants respond to the questions on the PHQ questionnaire. Recruiting patients in a more stable environment might give a totally different results.

6. PLOS authors have the option to publish the peer review history of their article (what does this mean?). If published, this will include your full peer review and any attached files.

Reviewer #1: No

Reviewer #2: **Yes: **Stewart Ndutard Ngasa

---

## [Author Response · Author response to Decision Letter 0]

24 Nov 2022

Response to Reviewers Comments

We express our gratitude to both reviewers for taking their time to review our manuscript and provide valuable feedback that has significantly improved our manuscript. We have endeavored to revise our manuscript thoroughly in line with reviewers’ suggestions. To this end, we have provided detailed responses to each comment raised by the reviewers and have mentioned the page and line numbers of changes in the manuscript. We have also incorporated reviewers’ suggestions and submitted the revised manuscript as cleaned and tracked versions.

Reviewer #1 comment:

1. I would suggest to scale the y-axis of figure 1 from 0-100.

Action: Figure 1 y-axis has been rescaled to 0-100

Reviewer #2 (Dr. Stewart Ndutard Ngasa) comments:

Comment 1

The article is entitled 'The Prevalence and Correlates of Depression among Patients with Chronic Diseases in the United Arab Emirates'. This suggests that all participants recruited have a chronic medical condition. In the manuscript, I couldn't find anywhere the definition for what was considered a chronic medical condition and how this guided the authors in choosing their participants. I would suggest including a section for inclusion and exclusion criteria. This will make it clearer for the readers.

Authors’ response: 

1- Thank you for the reviewer for pointing out to include a section on inclusion and exclusion criteria and a definition of a chronic medical condition.

Actions: 

We have included the definition in the introduction (Page: 4, Line: 100-103). Although we had an inclusion under the section of participants and sample size line 145 we have elaborated more in that section on inclusion criteria. (Page:6, Line :147-148)

Comment 2

2. I also found the analysis presented in table 2 a bit confusing, can the authors confirm that everyone who was included in the study had a chronic medical condition? If that was the case then our input variable can be broadly classified into 'present or absent' of a chronic medical condition and the outcome variables will be the correlates or associated factors. A separate table may be used to show the prevalence of depression in the different categories of chronic medical condition present in the participants

Authors’ response: 

2- Yes, all participants in the study had a chronic medical condition.

In Table 2 we studied the association between depression severity (outcome variable) and different demographic and comorbidities characteristics (covariate/factor variables). In this table, categorical variables were analyzed using Chi-square or Fisher’s exact test and continuous variable by Kruskal-Wallis test. Represents a medical condition (e.g. Type I diabetes) with one row (Present or Absent) instead of two rows (No & Yes) will only reduce the number of rows (i.e. table size) since eventually, we will use the Chi-square or Fisher’s exact test to get a p-value to decide if there is a significant association or not between the medical condition and depression severity.

3- We thank the reviewer for this suggestion.

Actions: 

1- To improve the understanding of Table 2. We restructured and moved the text describing Table 2 into a separate paragraph, updated the table’s title and added a footnote (Page: 9, Line: 216-220).

2- We added Fig 2 with the prevalence of depression in selected chronic diseases; diseases with adequate representation in the sample to provide an acceptable estimation of prevalence (Page: 8, Line: 206-207).

Comment 3

3. The authors have suggested in the paragraph following table 2 that the authors are describing a univariate result analysis. This seems to be the results of a bivariate analysis with Crude Odds Ratios presented. Can the authors verify this.

Authors’ response

We carried out a univariate logistic regression analysis, in which we examined the association of different characteristics with the presence of depression separately, this means that we fitted many models in each model we had one variable (e.g. age) and the outcome variable (depression). Please note that Tsai in his paper “Achieving consensus on terminology describing multivariable analyses” stated that a regression model studying the effect of single explanatory variable on the response variable is a univariable analysis. Using the term “bivariate” to describe such a model, while common, introduces unnecessary confusion [1].

Action: To avoid confusion, we replaced univariate and multivariable words in the manuscript with unadjusted/simple and adjusted/multiple logistic regression analysis, respectively (Page: 7, Line: 192, 196, 198; Page: 11, Line: 225, 230).

Comment 4

4. I was wondering if there was any rational for choosing the Ambulatory Health Services for the recruitment of patients, knowing that patients who present to such services usually come in an acute state. This might influence the way participants respond to the questions on the PHQ questionnaire. Recruiting patients in a more stable environment might give a totally different results.

Authors’ response: 

We agree with the reviewer. However, we chose it as the AHS clinics as those clinics provide the service of diagnosis, treatment, and follow-up with patients with chronic diseases. They have a special clinic for patients with chronic diseases (called CDC clinic). The choice was based on easy access to these patients and to raise awareness among the AHS staffs about the need to follow up on mental health of these patients. I would like to point that all patients receive the PHQ9 as part of the assessment, however, there are no data currently on the prevalence and the needs of mental health support for these patients.

Actions: 

We have added an explanation in (Page:5, Line:132-140) for the choice of AHS

References

[1] Tsai AC. Achieving consensus on terminology describing multivariable analyses. Am J Public Health. 2013 Jun;103(6):e1. doi: 10.2105/AJPH.2013.301234. Epub 2013 Apr 18. PMID: 23597350; PMCID: PMC3679183.

Editors Comments and Journal Requirements:

Response: Please note that the manuscript was edited as per journal requirement

Response: Please note that a statement on the consent was added in page 6, line 162-166

Response: We thank the editor for pointing out this issue. 

This study was funded by SUREPLUS Grant, G00003183 from United Arab Emirates University to be added in funding information

"This study was funded by SUREPLUS Grant, G00003183 from United Arab Emirates University."

"This study was funded by SUREPLUS Grant, G00003183 from United Arab Emirates University. The funders had no role in study design, data collection and analysis, decision to publish, or preparation of the manuscript."

Response: Please note that we removed information on funding from manuscript. Please add the following statement to the funding information

"This study was funded by SUREPLUS Grant, G00003183 from United Arab Emirates University. The funders had no role in study design, data collection and analysis, decision to publish, or preparation of the manuscript."

Response: we have uploaded the anonymized data set to a public repository Harvard Dataverse, available at the following DOI https://doi.org/10.7910/DVN/NRJNEN.

---

## [Decision Letter · Decision Letter 1]

28 Nov 2022

The Prevalence and Correlates of Depression among Patients with Chronic Diseases in the United Arab Emirates

PONE-D-22-26056R1

Dear Dr. Suliman,

We’re pleased to inform you that your manuscript has been judged scientifically suitable for publication and will be formally accepted for publication once it meets all outstanding technical requirements.

Kind regards,

Nabeel Al-Yateem, PhD

Academic Editor

PLOS ONE

Additional Editor Comments (optional):

Reviewers' comments:

Reviewer's Responses to Questions

**Comments to the Author**

1. If the authors have adequately addressed your comments raised in a previous round of review and you feel that this manuscript is now acceptable for publication, you may indicate that here to bypass the “Comments to the Author” section, enter your conflict of interest statement in the “Confidential to Editor” section, and submit your "Accept" recommendation.

Reviewer #1: All comments have been addressed

Reviewer #2: All comments have been addressed

2. Is the manuscript technically sound, and do the data support the conclusions?

Reviewer #1: Yes

Reviewer #2: Yes

3. Has the statistical analysis been performed appropriately and rigorously? 

Reviewer #1: Yes

Reviewer #2: Yes

4. Have the authors made all data underlying the findings in their manuscript fully available?

Reviewer #1: Yes

Reviewer #2: No

5. Is the manuscript presented in an intelligible fashion and written in standard English?

Reviewer #1: Yes

Reviewer #2: Yes

6. Review Comments to the Author

Reviewer #1: (No Response)

Reviewer #2: All suggestions for improvement have been made by the authors. This article now reads better and I recommend its accepted.

7. PLOS authors have the option to publish the peer review history of their article (what does this mean?). If published, this will include your full peer review and any attached files.

Reviewer #1: No

Reviewer #2: **Yes: **Stewart Ndutard Ngasa

---

## [Editor Report · Acceptance letter]

5 Dec 2022

PONE-D-22-26056R1 

The prevalence and correlates of depression among patients with chronic diseases in the United Arab Emirates 

Dear Dr. Suliman:

I'm pleased to inform you that your manuscript has been deemed suitable for publication in PLOS ONE. Congratulations! Your manuscript is now with our production department. 

Kind regards, 

on behalf of

Dr. Nabeel Al-Yateem 

Academic Editor

PLOS ONE